# Higher CCL22+ Cell Infiltration is Associated with Poor Prognosis in Cervical Cancer Patients

**DOI:** 10.3390/cancers11122004

**Published:** 2019-12-12

**Authors:** Qun Wang, Elisa Schmoeckel, Bernd P. Kost, Christina Kuhn, Aurelia Vattai, Theresa Vilsmaier, Sven Mahner, Doris Mayr, Udo Jeschke, Helene Hildegard Heidegger

**Affiliations:** 1Department of Obstetrics and Gynecology, University Hospital, LMU Munich, 80377 Munich, Germany; wqyxdz888@163.com (Q.W.); bernd.kost@med.uni-muenchen.de (B.P.K.); chrinstina.kuhn@med.uni-muenchen.de (C.K.); aurelia.vattai@med.uni-muenchen.de (A.V.); Theresa.vilsmaier@med.uni-muenchen.de (T.V.); sven.mahner@med.uni-muenchen.de (S.M.); Helene.heidegger@med.uni-muenchen.de (H.H.H.); 2Department of Pathology, LMU Munich, 80377 Munich, Germany; elisa.schmoeckel@med.uni-muenchen.de (E.S.); doris.mayr@med.uni-muenchen.de (D.M.)

**Keywords:** CCL22, FOXP3, cervical cancer, macrophage, T-reg

## Abstract

The chemokine CCL22 recruits regulatory T (T-reg) cells into tumor tissues and is expressed in many human tumors. However, the prognostic role of CCL22 in cervical cancer (CC) has not been determined. This study retrospectively analyzed the clinical significance of the expression of CCL22 and FOXP3 in 230 cervical cancer patients. Immunohistochemical staining analyses of CCL22 and FOXP3 were performed with a tissue microarray. Double immunofluorescence staining, cell coculture, and ELISA were used to determine CCL22 expressing cells and mechanisms. The higher number of infiltrating CCL22+ cells (CCL22^high^) group was associated with lymph node metastasis (*p* = 0.004), Fédération Internationale de Gynécologie et d’Obstétrique (FIGO) stages (*p* = 0.010), therapeutic strategies (*p* = 0.007), and survival status (*p* = 0.002). The number of infiltrating CCL22+ cells was positively correlated with that of infiltrating FOXP3+ cells (r = 0.210, *p* = 0.001). The CCL22^high^ group had a lower overall survival rate (OS), compared to the CCL22^low^ group (*p* = 0.001). However, no significant differences in progression free survival (PFS) were noted between the two groups. CCL22^high^ was an independent predictor of shorter OS (HR, 4.985; *p* = 0.0001). The OS of the combination group CCL22^high^FOXP3^high^ was significantly lower than that of the combination group CCL22^low^FOXP3^low^ regardless of the FIGO stage and disease subtype. CCL22^high^FOXP3^high^ was an independent indictor of shorter OS (HR, 5.284; *p* = 0.009). The PFS of group CCL22^high^FOXP3^high^ was significantly lower than that of group CCL22^low^FOXP3^low^ in cervical adenocarcinoma, but CCL22^high^FOXP3^high^ was not an independent indicator (HR, 3.018; *p* = 0.068). CCL22 was primarily expressed in M2-like macrophages in CC and induced by cervical cancer cells. The findings of our study indicate that cervical cancer patients with elevated CCL22+ infiltrating cells require more aggressive treatment. Moreover, the results provide a basis for subsequent, comprehensive studies to advance the design of immunotherapy for cervical cancer.

## 1. Introduction

Cervical cancer is the second most prevalent tumor in developing countries and the fourth most common cause of cancer-related deaths among women. Over half a million new cervical cancer cases, and an estimated 265,700 deaths are reported each year worldwide [1,2]. Several factors including economic conditions, genetic factors, endocrine [3], and immunity play significant role in the progression of cervical cancer. High-risk human papilloma virus is the primary cause of cervical cancer (CC) [4]. Immunosuppression states like infection with HIV [5] or taking immunosuppressive drugs [6] increases susceptibility to HPV infection and which subsequently causes cervical cancer. The current cervical cancer treatments include surgery, chemotherapy, and radiotherapy, but these are not effective for the management of advanced local cervical cancer, metastatic and recurrent tumors [7]. Recently, immunotherapy, particularly the immune checkpoint inhibitors, has achieved a great breakthrough. For example, the use of Pembrolizumab, a PD-1 inhibitor, in the later-staged and recurrent CC was authorized [8]. However, not all CC patients are sensitive to Pembrolizumab. Additional immune-related molecules should, therefore, be identified to explain the pathogenesis and improve the treatment of cervical cancer.

The role and mechanism of immune cells in the development of cervical cancer has not been adequately studied. Previous studies showed that the number of FOXP3+ regulatory T-cells that could suppress the innate and adaptive immunity systems [9,10] was higher in cervical cancer compared to other types of tumors [11] and suppressed immune responses [12]. On the one hand, it was reported that cervical cancer cells could secrete indoleamine 2,3-dioxygenase (IDO) to recruit FOXP3+ regulatory T-cells [12]. On the other hand, cervical cancer cells can secrete a series of molecules such as PEG2, IL-6, CCL2, and IL-10 that differentiate and activate M2-like macrophages [13,14,15,16]. M2-like macrophages have been widely accepted to play a role in the poor prognostic effect in CC [17]. M2-like macrophages promoted the CC cell proliferation by the GM-CSF/HB-EGF paracrine loop [18]. The number of M2-like macrophages was related to invasion patterns [19] and lymph node metastasis [20]. M2-like macrophages participate in immune suppression in cancer [21], yet, its underlying mechanism in CC has not been sufficiently elucidated. Previous studies on cervical cancer indicated that M2-like macrophages could decrease the presence of HPV16 E7 specific CD8+ T cells by diminishing HLA-DR expression and increasing the expression of either IL-10 or CSF1R [22,23,24]. Activated macrophages may inhibit the number of CD4+ T cells by producing neopterin [25]. M2-like macrophages could decrease the percentage of HPV specific regulatory T cells by blocking IL-10 signaling [26]. However, the association of CCL22 from cervical cancer cells or macrophages and regulatory T cells is still elusive.

The C-C motif chemokine ligand 22 (*CCL22*) gene is a secreted protein that exerts chemotactic activity for monocytes, dendritic cells, natural killer cells, and for chronically activated T lymphocytes [27,28,29,30]. Accumulating studies indicate that CCL22 plays a tumor-promoting role in human cancer. In ovarian cancer, for instance, CCL22 was found to induce regulatory T (T-reg) cells into tumor mass and inhibit T cell immunity [31]. High expression of CCL22 in M2 macrophages confers resistance to 5-fluorouracil in colorectal cancer [32]. A previous study showed that the *CCL22* mRNA expression level was higher in CC tissue than in a normal cervix [33]. However, the function of CCL22 in cervical cancer remains unknown.

The present study determined the functional role of CCL22 in infiltrating macrophages in cervical cancer. The expression level of CCL22 and the FOXP3+ regulatory T-cell marker was measured using a tissue microarray (TMA) with immunohistochemical staining. We further evaluated the correlation between clinical characteristics and CCL22 and FOXP3 expression. The findings of our study indicated that the number of CCL22+ cells was positively correlated with that of FOXP3+ cells (r = 0.210, *p* = 0.001). Moreover, group CCL22^high^ had a significantly lower overall survival rate (OS), compared to the CCL22^low^ group (*p* = 0.001). There was, however, no significant difference in progression free survival (PFS). The OS of the combination group CCL22^high^FOXP3^high^ was significantly lower than that of group CCL22^low^FOXP3^low^ regardless of the FIGO stage and disease subtype (*p* < 0.05). The PFS of group CCL22^high^FOXP3^high^ was significantly lower than that of group CCL22^low^FOXP3^low^ in cervical adenocarcinoma (*p* < 0.05). A double immunofluorescence staining indicated that M2-like macrophages primarily secreted CCL22. These results suggest that CCL22 secreted by M2 macrophages could recruit T-reg cells in cervical cancer and reduce the patient survival rate.

## 2. Results

### 2.1. CCL22 Was Overexpressed in Cervical Squamous Cell Carcinoma and Endocervical Adenocarcinoma (CESC)

The GEPIA database was used to identify the expression profile of CCL22 (http://gepia.cancer-pku.cn/) [34]. Transcript expression analysis for CCL22 was carried out in a total of 319 samples including 13 normal and 306 CESC tissues across TCGA normal and GTEx data. Next, using the ANOVA method, 1 as the Log2FC cutoff value, 0.01 as the cutoff value of the significance level, the expression difference of CCL22 in CESC tissue was obtained (Figure 1). The CCL22 mRNA level in CESC tissue was much higher than that in normal cervical tissues.

TIMER database was also used to identify the correlation between T-regs, TAM2, and CCL22 (https://cistrome.shinyapps.io/timer/) [35]. MRC1 (also named CD206) and FOXP3 represent M2 macrophage and regulatory T-cells, respectively [12,36]. Correlation analysis for MRC1, FOXP3, and CCL22 was carried out in the CESC dataset from TCGA. The results showed that CCL22 was positively correlated with MRC1 and FOXP3 (r = 0.329, *p* = 4.45 × 10^−9^; r = 0.385, *p* = 4.31 × 10^−12^, respectively). MRC1 was positively correlated with FOXP3 (r = 0.43, *p* = 0.001). The UALCAN database was used to analyze the survival rate in groups with differently expressed CCL22 in the CESC dataset from TCGA (http://ualcan.path.uab.edu/index.html) [37]. The result showed that although there was no significant difference, and the OS of the high CCL22 expression group was lower than that of the low CCL22 expression group in the long run (*p* = 0.069) (Appendix A).

### 2.2. The Association between the IRS of CCL22 in CC Cells, Infiltrating CCL22+ Cell and FOXP3+ Cell Counts with Clinical Characteristics

Tissue microarray by immunohistochemistry was performed to test the number of CCL22+ and FOXP3+ cells in a retrospective cohort of 230 cervical cancer cases, including 187 cases of squamous carcinoma and 43 cases of adenocarcinoma. As shown in Figure 2 and Figure 3, the immunoreactivity of CCL22 and FOXP3 protein were detected. The IRS of CCL22 expressed in cervical cancer cells was evaluated and the number of CCL22+ and FOXP3+ cells was counted. We found that CCL22 expressed in CC cells was significantly associated with the disease subtypes (*p* < 0.05). The higher number of CCL22+ cells was significantly associated with lymph node metastasis (*p* < 0.05), FIGO stages (*p* < 0.05), therapeutic strategies (*p* < 0.05), and survival status (*p* < 0.05). The group with lower number of CCL22 (+) cells had 42.7% (88/206) lymph node metastasis, 55.8% (115/206) FIGO stage I to II, 38.3% (79/206) cases treated by surgery, and 11.2% (23/206) in death. The group with the higher number (*n* > 11) of CCL22 (+) cells had 12.5% (3/24) of cancer with lymph node metastasis, 83.3% (20/24) FIGO stage I to II, 70.8% (17/24) cases treated by surgery, and 37.5% (9/24) of cancer with death. The higher number of FOXP3+ was significantly associated with grading (*p* < 0.05). The number of FOXP3+ cells was significantly associated with disease subtypes (*p* < 0.05). The higher number (*n* > 29) of FOXP3+ cells was noted in 94.1% (32/34) with squamous carcinoma, while a lower number of FOXP3+ cells was observed in 79.1% (155/196) with squamous carcinoma (Table 1).

### 2.3. Higher Number of CCL22+ Cells Predicts Poor Prognosis of Patients with Cervical Cancer

We compared the overall survival rate (OS) and progression free survival (PFS) between group CCL22^low^ and group CCL22^high^ and the combined groups of CCL22 and FOXP3. The results of this study revealed that the OS of group CCL22^high^ was significantly lower than that of group CCL22^low^ (*p* = 0.001) (Figure 2A, left). The OS of group CCL22^high^FOXP3^low^ was significantly lower than that of group CCL22^low^FOXP3^low^ (*p* = 0.022), while the OS of group CCL22^high^FOXP3^high^ was significantly lower than that of group CCL22^low^FOXP3^low^ (*p* = 0.003). The OS of group CCL22^high^ FOXP3^high^ was significantly lower than that of group CCL22^low^ FOXP3^high^ (*p* = 0.019) (Figure 2B, left, Table 2). There was no significant difference in PFS between group CCL22^low^ and group CCL22^high^ (Figure 2A, right), but PFS of group CCL22^high^FOXP3^high^ was significantly lower than that of group CCL22^low^FOXP3^high^ (Figure 2B, right, Table 3).

We further compared the OS and PFS of the four groups in low and high FIGO stages, respectively. We found that the OS of CCL22^high^FOXP3^low^ and CCL22^high^FOXP3^high^ groups were significantly lower than those of the CCL22^low^FOXP3^low^ group in FIGO stage I–IIA (*p* = 0.001, *p* = 0.013, respectively) (Figure 2C, left, Table 4). The OS of the CCL22^high^FOXP3^high^ group was significantly lower than that of the CCL22^low^FOXP3^low^ group in FIGO stage IIB–IV (*p* = 0.029) (Figure 2C, right, Table 4). However, there was no significant difference in PFS among different groups in both low and high FIGO stages (*p* > 0.05) (Figure 2D, Table 5).

Due to the significant difference in the distribution of FOXP3(+) cells in cervical squamous carcinoma and adenocarcinoma (Table 1), we further compared the OS and PFS of the four groups in the two disease types, respectively. We found that the OS of the CCL22^high^FOXP3^high^ group was significantly lower than that of the CCL22^low^FOXP3^high^ and CCL22^low^FOXP3^low^ groups in cervical squamous carcinoma (*p* = 0.023, *p* = 0.020, respectively) (Figure 2E, left, Table 6). The OS of group CCL22^high^FOXP3^high^ was significantly lower than that of group CCL22^low^ FOXP3^low^ in cervical adenocarcinoma (*p* = 0.043) (Figure 2E, right, Table 6). There was no significant difference in PFS in cervical squamous carcinoma (*p* > 0.05) (Figure 2F, left, Table 7). The PFS of group CCL22^high^FOXP3^high^ was significantly lower than that of group CCL22^low^FOXP3^low^ in cervical adenocarcinoma (*p* = 0.041) (Figure 2F, right, Table 7).

Subsequently, a multivariate analysis was done using the COX proportional hazards model for variables that were significant in the univariate analysis was performed. For the OS analysis, we constructed two models with CCL22^low^ vs. CCL22^high^ or with CCL22^low^FOXP3^low^ vs. CCL22^high^FOXP3^high^. For the PFS, one model with CCL22^low^FOXP3^low^ vs. CCL22^high^FOXP3^high^ was constructed. The results showed that CCL22^high^ and CCL22^high^FOXP3^high^ were independent predictors of shorter OS (HR 4.985, *p* = 0.0001; HR 5.284, *p* = 0.009; respectively). Although there were significant differences in PFS between group CCL22^high^FOXP3^high^ and group CCL22^low^FOXP3^low^, CCL22^high^FOXP3^high^ was not an independent predictor of PFS (HR, 3.018; *p* = 0.068) (Table 8).

### 2.4. Correlation Analysis between CCL22+ Expression and FOXP3+ Cells

FoxP3 has been employed as a marker of regulatory T-cells. We postulated that the number of CCL22+ cells or the CCL22 expression in cervical cancer cells is correlated to the number of FOXP3+ cells in cervical cancer specimens. This hypothesis was tested by determining the expression of CCL22 and FOXP3 proteins by tissue microarray with IHC (Figure 3). A spearman correlation analysis indicated that both the number of CCL22+ cells and the CCL22 expression in cervical cancer cells were positively correlated with that of FOXP3+ cells (r = 0.210, *p* = 0.002; r = 0.144, *p* = 0.027, respectively) (Figure 4).

### 2.5. Identification of CCL22 Expressing Cells in Cervical Cancer

Placenta tissue was used as a positive control for CCL22 and CD68, and appendix tissue as a positive control tissue for CD163. Negative control regents with normal IgG were applied to check CCL22 (rabbit IgG), CD68 (mouse IgG), and CD163 (mouse IgG) antibody specificity (Figure 5 and Figure 6).

A previous study reported that CCL22 is expressed in myeloid cells such as dendritic cells and macrophages in the steady state [38]. This study, therefore, performed double immunofluorescence staining for CCL22, CD68, and CD163 to ascertain the expression of CCL22 in CC macrophages. CD68 and CD163 are specific markers for macrophages and M2-like macrophages, respectively [39]. We analyzed the percentage of CD68+/CCL22-, CD68+/CCL22+ and CD163+/CCL22-, CD163+/CCL22+ cells with eight slides. Our results showed that 55.22% (±14.32%) CCL22+CD68+ cells were found on CD68+ cells, but 82.55% (±22.23%) CCL22+CD163+ cells were found on CD163+ cells (*p* = 0.014, Student’s *t* test). The results of our study indicate that M2-like macrophages primarily secrete CCL22.

### 2.6. Cervical Cancer Cells Induced CCL22 in Monocytes

We cocultured monocytes with cervical cancer cells in a noncontact transwell system, which allowed the exchange of soluble factors, but prevented direct cell-cell contact. After coculture, ELISA assay demonstrated that the concentration of CCL22 in the supernatant of the lower chamber of cocultures was significantly higher (1405.4 ± 15.8, *p* < 0.05; 293.2 ± 13.3, *p* < 0.05; with Caski cells and Hela cells, respectively) than that of monocytes (0.000007 ± 0.0000004) or cervical cancer cells alone (73.3 ± 13.7, *p* < 0.05; 10.4 ± 18.1, *p* < 0.05; Caski and HeLa cells, respectively) (Figure 7). These results indicated that cervical cancer cells could induce CCL22 in monocytes.

## 3. Materials and Methods

### 3.1. Bioinformatics

The GEPIA database was used to identify the expression profile of CCL22 (http://gepia.cancer-pku.cn/) [34]. Transcript expression analysis for CCL22 was carried out in a total of 316 samples including 13 normal and 303 CESC tissues across TCGA normal and GTEx data. Next, using ANOVA method, 1 as the Log2FC cutoff value, 0.01 as the cutoff value of the significance level, the expressing difference of CCL22 in CESC tissue was obtained. TIMER database was used to identify the correlation between T-regs, TAM2, and CCL22 (https://cistrome.shinyapps.io/timer/) [35]. UALCAN database was used to analyze survival rate in groups with differently expressed CCL22 in CESC dataset from TCGA (http://ualcan.path.uab.edu/index.html) [37].

### 3.2. Clinical Sample

The Institutional Review Board of the Ludwig-Maximilian University, Munich, (Number of approval: 337-06, 29 December 2006) approved this study. All patients provided written informed consent. Human cervical cancer tissue microarrays contained 230 samples from the patients diagnosed with CC between January 1994 and September 2002 consecutively from the Department of Obstetrics and Gynecology, LMU Munich. The age was from 22 to 83 years with a median age of 49 years. Clinicopathologic characteristics of patients are available in Table 1. Staging was performed according to the FIGO staging system. Primary treatment of CC consisted of type 3 radical hysterectomy with pelvic lymph-node dissection. Data including age, stage, lymph node metastasis, distant metastasis, and survival status were obtained from medical records. Data on tumor size, tumor grade, and lymph-node metastases were reviewed from pathology reports.

### 3.3. Tissue Microarray Construction

TMAs were constructed from 230 formalin-fixed, paraffin-embedded cervical cancer tissue specimens, including 187 cervical squamous cancer and 43 cervical adenocarcinoma. Briefly, to define representative tumor areas, an institutional pathologist reviewed the hematoxylin and eosin (HE)-stained whole-mount sections to define representative tumor areas. Subsequently, after retrieval of four 1.0-mm diameter tissue cores from the formalin-fixed paraffin-embedded tissue blocks of which consisted tumor specimens, they were arrayed on a 38 × 25 recipient paraffin block using an MTA-1 manual tissue array (Beecher Instruments Inc., Silver Spring, MD, USA). Then, 3 μm sections were cut on a microtome and placed on glass slides for two replicates. One series was for staining CCL22 and the other was for staining FoxP3. The presence of tumor tissue on the sections was verified by HE staining.

### 3.4. Immunohistochemistry

The TMA sections were deparaffinized in xylol for 20 min and rinsed in 100% ethanol. Then, 20 min H_2_O_2_ incubation was performed to inhibit endogenous peroxidase reaction. Afterwards, the specimens were rehydrated in deescalating alcohol gradients, starting with 100% ethanol and ending with distilled water. The pressure pot contained a sodium citrate buffer (pH = 6.0), which consisted of 0.1 m citric acid and 0.1 mM sodium citrate in distilled water. Then, samples were washed in PBS twice and incubated with a blocking solution (Regent 1, ZytoChem Plus HRP Polymer System (Mouse/Rabbit), Zytomed, Berlin, Germany) for 5 min. Incubation with the primary antibody was performed at 4 °C for 16 h. All antibodies used are listed in Table 4. Samples were washed twice in PBS (pH = 7.4) following every subsequent step. The blocking solutions (Regent 2) was applied for 20 min and HRP-Polymer (Regent 3) for 30 min. The chromogen-substrate staining was carried out using the Liquid DAB+ Substrate Chromogen System (Dako Scientifi, Glostrup, Denmark). The reaction was stopped by applying distilled water. Finally, the slides were counterstained with Hemalaun for 2 min and blued in tap water. The slides were dehydrated in an ascending alcohol gradient and cover slipped with Eukitt quick hardening mounting medium (Sigma Aldrich, St.Lousis, MO, USA). Human tonsil tissue was applied as positive control. All slides were analyzed using the microscope Leitz Wetzlar (Wetzlar, Germany; Type 307-148.001 514686). The immunoreactive score (IRS) was used for evaluation of the intensity and distribution pattern of antigen expression. The semiquantitative score was calculated as follows: the optical staining intensity (grades: 0 = none, 1 = weak, 2 = moderate, 3 = strong staining) was multiplied by the total percentage of positively stained cells (0 = none, 1 = 10%, 2 = 11–50%, 3 = 51–80%, and 4 = 81% of the cells). This multiplication has a minimum of 0 and a maximum of 12. Two experience staff members analyzed the slides independently. Total number of CCL22+ cells and FOXP3+ cells were counted in a magnification field of 40× lens three times each in three different areas of each slide. The cutoff value was calculated by ROC curve. Cutoff values of 11 and 27 were applied for CCL22+ cells and FOXP3+ cells, respectively.

#### 3.4.1. Evaluation of CCL22+ Cells as Macrophages

For the visualization of CCL22 expressing cells in CC infiltrating cells, the same tissue samples with TMA were used. The antibodies are shown in Table 4. Double immunofluorescence staining for CCL22 and CD68 as a specific macrophage marker was performed to identify the CCL22 expression in macrophages.

#### 3.4.2. Evaluation of CCL22+ Cells as M2-Like Macrophages

In order to identify which subtype of macrophages express CCL22, the same tissue samples with TMA were used. The antibodies are shown in Table 9. Double immunofluorescence staining for CCL22 and CD163 as a specific M2-like macrophage marker was performed.

### 3.5. Cell Coculture

To determine the effect of cervical cancer cells on the induction of CCL22 in macrophages, monocytes were cocultured with HeLa cells and Caski cells, respectively. Transwell core size was 0.4 μm. First, we picked up the ordered buffy coat and purified the PBMCs using the Ficoll–Hypaque method. We counted the PBMCs and seeded 0.2 million PBMCs per well into 24-well plates. Then plastic adherence was in the incubator overnight. The next day, adhered monocytes were washed gently with 1 × PBS, digested by 0.25% trypsin with 0.25% EDTA and ended by complete medium. Monocytes were plated in the lower chamber and 10^7^ cervical cells were plated in the upper chamber in 24 wells. They were cultured in 500 μL culture medium in each chamber with RPMI 1640 medium and 10% fetal calf serum. Cocultures were incubated for 48 h and the supernatant of the lower chambers were harvested followed by Elisa. 0.2 million monocytes with inserts and 10^7^ Caski and 10^7^ HeLa cells with inserts cultured with RPMI 1640 medium with 10% fetal cow serum were as the control. Elisa was performed after 48 h.

### 3.6. Elisa

Serum of all groups were collected and stored in −80 °C before analysis. ELISA assay was performed according to the manufacturer’s instructions of the ELISA kits (Catalog No. DMD00, R&D Systems, Minneapolis, MN, USA). The OD value at 450 nm was measured. The concentrations of CCL22 was calculated according to the standard curve.

### 3.7. Statistical Analysis

Clinical data are presented as number (percentage of the amount of relevant clinicopathology). Statistical analysis of the number of CCL22+ and FOXP3+ cells were performed using chi-square test. Correlation analysis of the number of CCL22+ and FOXP3+ cells were performed using Spearman correlation. The Kaplan–Meier method was used to calculate the OS curve and further survival analysis was performed using the log-rank test. The Cox proportional hazard model was used to estimate hazard ratios and 95% confidence intervals in both univariate and multivariate models. CCL22 concentration was presented as mean ± standard deviation. ANOVA analysis was performed using Welch ANOVA and Games-Howell. Statistical analyses were performed using SPSS version 23.0 (SPSS Inc., Chicago, IL, USA). *p* < 0.05 was considered statically significant.

## 4. Discussion

Our study revealed a correlation between the expression of CCL22 and FOXP3 in cervical cancer. This study is the first to report the relationship between CCL22 expression and the prognosis of cervical cancer (CC) patients. The group with a higher number of CCL22+ infiltrating cells shows a lower overall survival rate (OS) than that of the group of lower number of CCL22+ infiltrating cells in cervical cancer. However, the expression of CCL22 in cervical cancer cells seems not to affect the survival rate of CC patients. Moreover, the OS of the group combining higher CCL22+ and FOXP3+ infiltrating cells is lower than that of the group with a combined lower number of CCL22+ and FOXP3+ infiltrating cells.

In recent years, several studies have investigated the prognostic value of new immune checkpoints such as CCL22. CCL22 belongs to a family of secreted proteins that play various roles in immunoregulatory and inflammatory processes [27]. Previous research has reported that CCL22 is a prognostic predictor of various cancers. For instance, a report by Li et al. indicated that tumor secretion by CCL22 is an independent prognostic predictor of breast cancer [40]. Another study proved that the levels of serum macrophage-derived CCL22 are associated with glioma risk and survival period [41]. Besides, CCL22 predicts postoperative prognosis in patients with stage II/III gastric cancer [42]. The prognostic value of CCL22 in cervical cancer has, however, not been investigated. This study reveals that the CCL22^high^ group had a lower OS, compared to the CCL22^low^ group (*p* = 0.001). CCL22^high^ was an independent predictor of shorter OS (HR, 4.985; *p* = 0.0001). The OS of the combination group CCL22^high^FOXP3^high^ was significantly lower than that of the combination group CCL22^low^FOXP3^low^ regardless of the FIGO stage and disease subtype. CCL22^high^FOXP3^high^ was an independent indictor of shorter OS (HR, 5.284; *p* = 0.009). The PFS of group CCL22^high^FOXP3^high^ was significantly lower than that of group CCL22^low^FOXP3^low^ in cervical adenocarcinoma, but CCL22^high^FOXP3^high^ was not an independent indicator (HR, 3.018; *p* = 0.068). Therefore, the CCL22+ infiltrating cells or the combination of CCL22 (+) and FOXP3 (+) cells are novel biomarkers that can be potentially used for cervical cancer prognosis. Since this study only had 5/230 samples with both a high number of CCL22+ and FOXP3+ infiltrating cells, increasing the sample size might provide a more reliable result.

Previous studies reported that CCL22 could recruit FOXP3 (+) regulatory T-cells. Increased CCL22 mRNA levels are correlated with increased FoxP3 mRNA levels in oral cancer specimens [43]. In a B16F10 melanoma model, imiquimod (IQM) has been shown to reduce T-regs at the tumor site by the downregulation of CCL22 production [44]. In cervical cancer, the *CCL22* mRNA levels of neoplastic foci and tumor periphery is positively correlated with FOXP3 [45]. In cervical cancer, however, the correlation between protein expression levels of CCL22 and FOXP3, and the cell source of CCL22 have not been determined. This study revealed a similar trend, that the protein level of CCL22 from both cervical cancer cells and infiltrating cells was positively correlated with FOXP3. This correlation was defined by calculating the IRS of CCL22 expression in cervical cancer cells and counting the number of CCL22+ and FOXP3+ cells. The association of the latter was better than the former. Therefore, both cervical cancer cell-derived and infiltrating cell-derived cells might recruit T-regs, and the role of CCL22 from infiltrating cells was found to be more significant than that of CCL22 from cervical cancer cells.

Several studies have investigated the source of CCL22. For instance, Huang et al. proved that CCL22 was overexpressed in head and neck cancer cells [43]. CCL22 was found in macrophages of tongue squamous cell carcinoma [46]. Additionally, the secretion of CCL22 by M2-like macrophages in colorectal cancer was proven [32]. However, CCL22 expression in cervical cancer has not been previously determined. By performing a tissue microarray (TMA) using immunohistochemistry (IHC), we showed that CCL22 could be secreted by cervical cancer cells. Besides, double immunofluorescence of CD163 and CCL22 showed that 82.55% (±22.23%) of CD163(+) cells were overlapped with CCL22 (+) infiltrating cells. Therefore, CCL22 is secreted by M2-like macrophages and cervical cancer cells in cervical cancer. Moreover, many studies proved that cervical cancer cells could induce monocytes into M2 macrophages [18,42] which was consistent with the result in our study to some degree that CCL22 in monocytes could be induced by cervical cancer cells. However, further studies need to be performed to show the effect of cervical cancer cells on M2 macrophages and the regulatory mechanisms of CCL22 needs to be studied in the future. In conclusion, our findings suggest that the increase in CCL22 (+) infiltrating M2-like macrophage cells may recruit more T-regs in CC tissue and cause a poor prognosis for cervical cancer patients. CCL22 could be a prognostic predictor and therapeutic target to identify and treat cervical cancer patients with poorer clinical outcomes.

## 5. Conclusions

Our study demonstrates that high CCL22(+) infiltrating cells particularly M2-like macrophage cells, is associated with a poor outcome of cervical cancer patients. CCL22 expression is positively correlated with FoxP3 expression in cervical cancer. Thus, CCL22 may be a novel prognostic marker and therapeutic target for the treatment of cervical cancer.

## Figures and Tables

**Figure 1 cancers-11-02004-f001:**
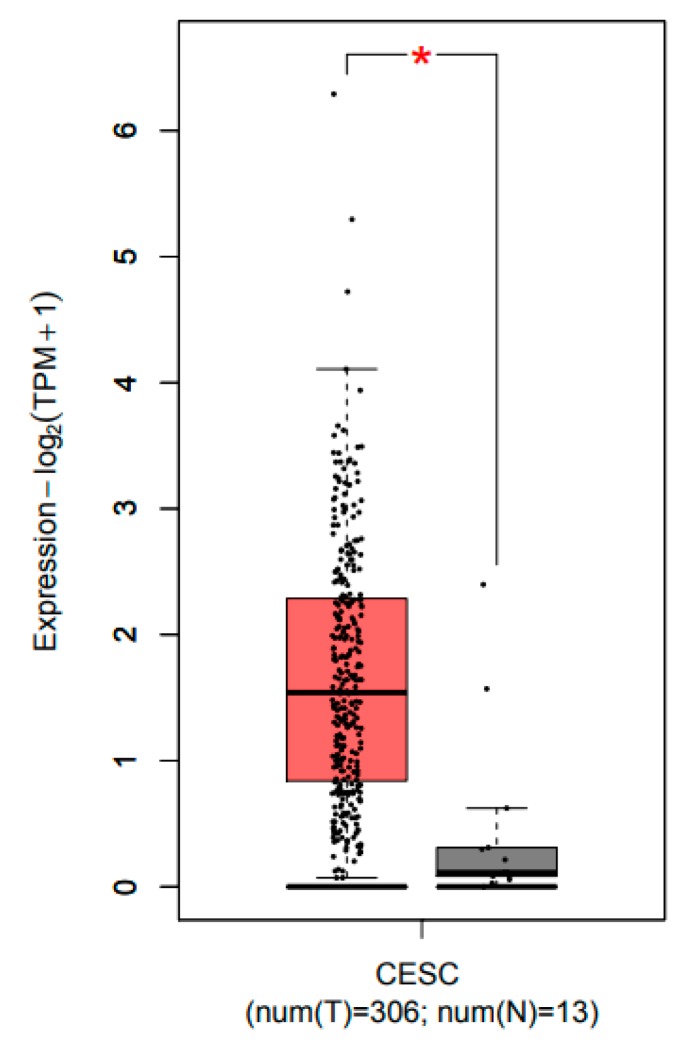
Transcripts expression level of CCL22 in CESC tissue explored using the GEPIA database. Red and grey colors denote the expression level in tumor tissue and normal tissue, respectively. CESC, cervical squamous cell carcinoma and endocervical adenocarcinoma. The asterisk (*) indicate significant higher CCL22 expression in tumor tissue compared to normal tissue.

**Figure 2 cancers-11-02004-f002:**
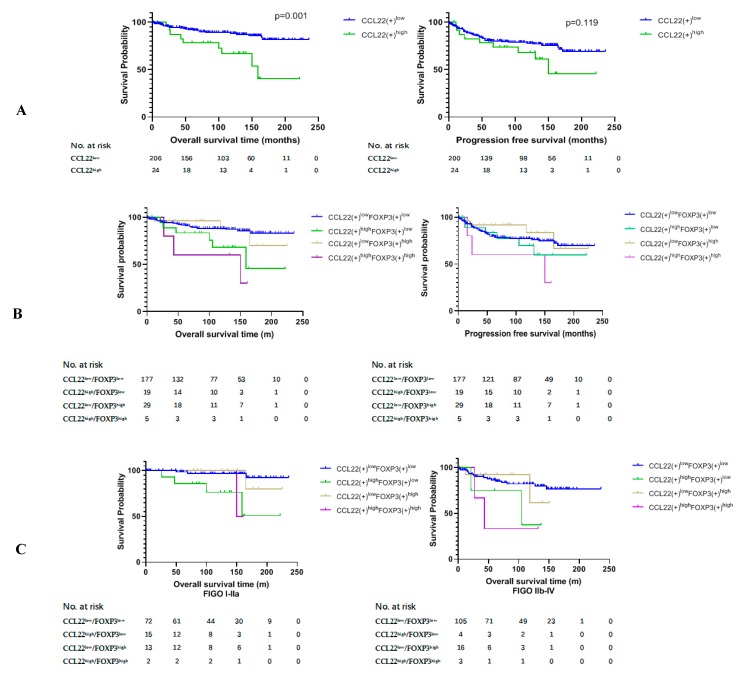
Kaplan–Meier estimates of overall survival rate and progression-free survival. (**A**) Overall survival rate (OS) (left) and progression free survival (PFS) (right) of group CCL22^low^ and CCL22^high^. (**B**) OS (left) and PFS (right) of the combined groups. (**C**) OS of the combined groups at different FIGO stages. (**D**) PFS of the combined groups at different FIGO stages. (**E**) OS of the combined groups in different disease subtypes. (**F**) PFS of the combined groups in different disease subtypes.

**Figure 3 cancers-11-02004-f003:**
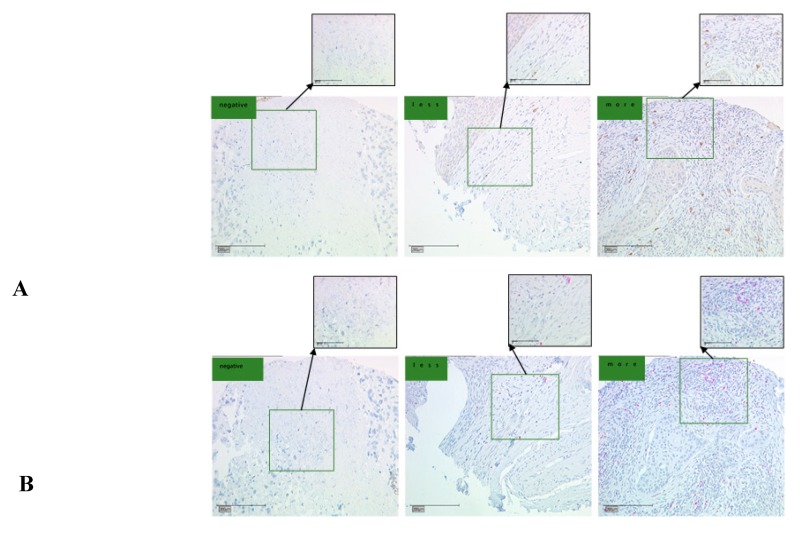
CCL22 and FOXP3 expression in cervical cancer (CC) tissue microarray as determined by IHC. (**A**) IHC for CCL22. The scale is 100 and 200 μm, respectively; (**B**) IHC for FOXP3. The scale is 100 and 200 μm, respectively. The number of CCL22+ and FOXP3+ cells was counted and classified as either negative, less, or more.

**Figure 4 cancers-11-02004-f004:**
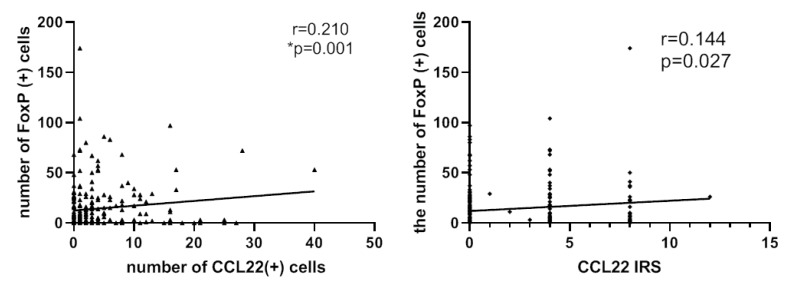
Left: number of CCL22+ cells is associated with that of FOXP3+ cells (r = 0.210, *p* = 0.001). Right: IRS of CCL22 expressed in cervical cancer cells is associated with the number of FOXP3+ cells (r = 0.144, *p* = 0.027). The number of cells was counted at a magnification of 25× lens and calculated three times each in three different areas of the tissue microarray (TMA). The IRS of CCL22 in cervical cancer cells was evaluated at a magnification of 25× and evaluated by two independent researchers.

**Figure 5 cancers-11-02004-f005:**
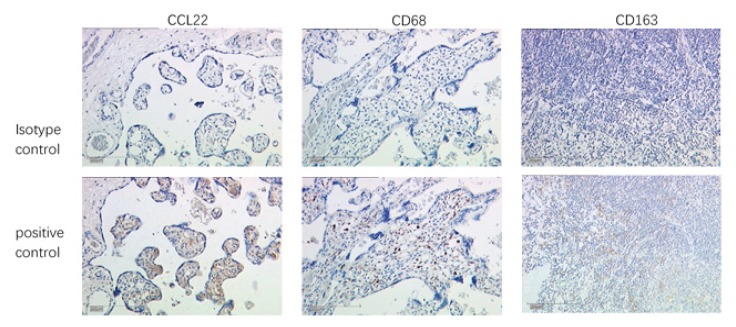
Validation of antibody specificity by IHC. Placenta as positive control tissue for CCL22 and CD68, and appendix as positive control tissue for CD163. The scale is 200 μm.

**Figure 6 cancers-11-02004-f006:**
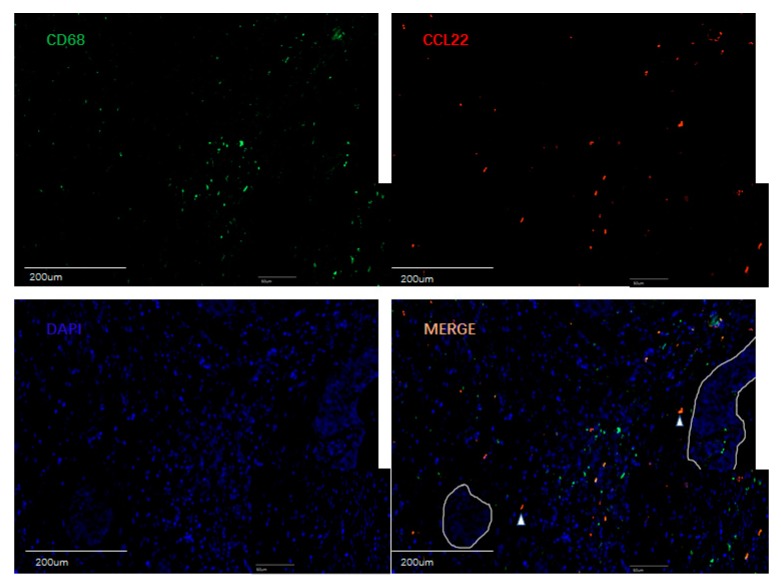
Double fluorescence staining of cervical cancer detecting CD68+CCL22+ cells and CD163+CCL22+cells. Immunofluorescent staining of CCL22+ cells (in red) expressing CD68 (macrophages in green) and CD163 (M2 macrophages in green). These two sets of pictures of different cervical tumor samples are representative of eight tumors analyzed. Arrowheads in the merged picture (lower right) designate double-positive cells located in the stroma. 55.22% (±14.32%) CCL22+CD68+ cells were found on CD68+ cells, but 82.55% (±22.23%) CCL22+CD163+ cells were found on CD163+ cells (*p* = 0.014, Student’s *t* test). The solid line in white indicates a barrier between stroma and tumor nest. The image was shown in original magnification of 10× and 40×.

**Figure 7 cancers-11-02004-f007:**
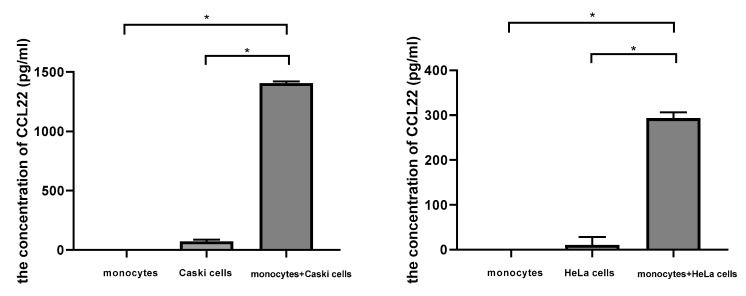
Cervical cancer cells induced CCL22 in monocytes. The left column graph representing the concentration of CCL22 in the supernatant from monocytes, Caski cells, and the cocultures of monocytes and Caski cells, respectively. The right column graph represents the concentration of CCL22 in the supernatant from monocytes, HeLa cells, and the cocultures of monocytes and HeLa cells, respectively. Each experiment was triplicated. * *p* < 0.05.

**Table 1 cancers-11-02004-t001:** The association between the number of CCL22+ cells and FOXP3+ cells and clinicopathological characteristics.

Clinicopathological Characteristics	Total (%)	IRS of CCL22 in CC Cells		*p*	Quantity of CCL22+ (n%)		*p*	Quantity of FOXP3+ (n%)		*p*
		Low (IRS < 4)	High (IRS ≥ 4)		Small (*n* ≤ 11)	Large (*n* > 11)		Small (*n* ≤ 29)	Large (*n* > 29)	
All cases	230	134 (58.3%)	96 (41.7%)	−	206 (89.6%)	24 (10.4%)	−	196 (85.2%)	34 (14.8%)	−
**Age (year)**										
≤50	132 (57.4%)	80 (59.7%)	52 (54.2%)	0.403	114 (55.3%)	18 (75.0%)	0.065	115 (58.7%)	17 (50.0%)	0.345
>50	98 (42.6%)	54 (40.3%)	44 (45.8%)		92 (44.7%)	6 (25.0%)		81 (41.3%)	17 (50.0%)	
**Tumor Size (cm)**										
<2	2 (0.9%)	2 (1.5%)	0 (0%)	0.080	2 (1.0%)	0 (0%)	0.090	2 (1.0%)	0 (0%)	0.905
2–4	120 (52.2%)	75 (56%)	45 (46.9%)		103 (50.0%)	17 (70.8%)		102 (52%)	18 (52.9%)	
>4	107 (46.5%)	57 (42.5%)	51 (53.1%)		101 (49.0%)	7 (29.2%)		92 (46.9%)	16 (47.1%)	
**PN**										
Without lymph node metastasis	139 (60.4%)	81 (60.4%)	58 (60.4%)	0.996	118 (57.3%)	21 (87.5%)	0.004 *	122 (62.2%)	17 (50.0%)	0.178
With lymph node metastasis	91 (39.6%)	53 (39.6%)	38 (39.6%)		88 (42.7%)	3 (12.5%)		74 (37.8%)	17 (50.0%)	
**PM**										
Without metastasis	219 (95.2%)	127 (94.8%)	92 (95.8%)	0.709	195 (94.7%)	24 (100%)	0.115	187 (95.4%)	32 (94.1%)	0.752
With metastasis	11 (4.8%)	7 (5.2%)	4 (4.2%)		11 (5.3%)	0 (0%)		9 (4.6%)	2 (5.9%)	
**FIGO**										
I–IIA	102 (44.3%)	62 (46.3%)	40 (41.7%)	0.488	85 (41.3%)	17 (70.8%)	0.006 *	87 (60.2%)	15 (50.0%)	0.265
IIB–IV	128 (55.7%)	55 (53.7%)	40 (58.3%)		121 (58.7%)	7 (29.2%)		109 (39.8%)	19 (50.0%)	
**Grade**										
1	19 (8.3%)	10 (7.8%)	9 (9.5%)	0.676	17 (8.9%)	2 (6.1%)	0.653	13 (6.7%)	4 (12.1%)	0.243
2	129 (56.1%)	72 (56.3%)	58 (60.0%)		112 (58.9%)	17 (51.5%)		123 (63.1%)	12 (36.4%)	
3	75 (32.6%)	46 (35.9%)	29 (30.5%)		61 (32.1%)	14 (42.4%)		59 (30.3%)	17 (51.50%)	
NA	7 (3.0%)									
**Subtype**										
Squamous carcinoma	187 (81.3%)	95 (70.9%)	92 (95.8%)	0.001 *	171 (83.0%)	16 (66.7%)	0.091	155 (79.1%)	32 (94.1%)	0.038 *
Adenocarcinoma	43 (18.7%)	39 (29.1%)	4 (4.2%)		35 (17.0%)	8 (33.3%)		41 (20.9%)	2 (5.9%)	
Progression										
No	175 (76.1%)	99 (75.0%)	76 (80.0%)	0.377	160 (78.8%)	15 (62.5%)	0.072	149 (76.8%)	26 (78.8%)	0.802
Yes	52 (22.6%)	33 (25.0%)	19 (20.0%)		43 (21.2%)	9 (37.5%)		45 (23.2%)	7 (21.2%)	
NA	3 (1.3%)									
**Therapeutic Strategy**										
Surgery	96 (41.7%)	59 (44.0%)	37 (38.5%)	0.675	79 (38.3%)	17 (70.8%)	0.007 *	83 (42.3%)	13 (38.2%)	0.494
Preoperative chemoradiotherapy	6 (2.6%)	3 (2.2%)	3 (3.1%)		6 (2.9%)	0 (0.0%)		4 (2.0%)	2 (5.9%)	
Chemoradiotherapy	128 (55.7%)	72 (53.7%)	56 (58.3%)		121 (58.7%)	7 (29.2%)		109 (55.6%)	19 (55.9%)	
**Survival State**										
Alive	198 (86.1%)	114 (85.1%)	84 (87.5%)	0.600	183 (88.8%)	15 (62.5%)	0.002 *	170 (86.7%)	28 (82.4%)	0.507
Death	32 (13.9%)	20 (14.9%)	12 (12.5%)		23 (11.2%)	9 (37.5%)		26 (13.3%)	6 (17.6%)	

NA, not applicable as data not available. * *p* < 0.05.

**Table 2 cancers-11-02004-t002:** Pairwise comparison of overall survival rate.

Group		CCL22^low^ FOXP3^low^		CCL22^high^ FOXP3^low^		CCL22^low^ FOXP3^high^		CCL22^high^ FOXP3^high^	
		X^2^	*p*-Value	X^2^	*p*-Value	X^2^	*p*-Value	X^2^	*p*-Value
Log Rank (Mantel–Cox)	CCL22^low^ FOXP3^low^	−	−	5.250	0.022 *	0.005	0.946	8.732	0.003 *
	CCL22^high^ FOXP3^low^	5.250	0.022 *	−	−	2.466	0.116	0.469	0.494
	CCL22^low^ FOXP3^high^	0.005	0.946	2.466	0.116	−	−	5.486	0.019 *
	CCL22^high^ FOXP3^high^	8.732	0.003 *	0.469	0.494	5.486	0.019 *	−	−

* *p* < 0.05.

**Table 3 cancers-11-02004-t003:** Pairwise comparison of progression-free survival.

Group		CCL22^low^ FOXP3^low^		CCL22^high^ FOXP3^low^		CCL22^low^ FOXP3^high^		CCL22^high^ FOXP3^high^	
		X^2^	*p*-Value	X^2^	*p*-Value	X^2^	*p*-Value	X^2^	*p*-Value
Log Rank (Mantel–Cox)	CCL22^low^ FOXP3^low^	−	−	0.519	0.471	0.490	0.484	3.320	0.068
	CCL22^high^ FOXP3^low^	0.519	0.471	−	−	1.431	0.232	0.353	0.552
	CCL22^low^ FOXP3^high^	0.490	0.484	1.431	0.232	−	−	4.069	0.044 *
	CCL22^high^ FOXP3^high^	3.320	0.068	0.353	0.552	4.069	0.044 *	−	−

* *p* < 0.05.

**Table 4 cancers-11-02004-t004:** Pairwise comparison of overall survival rate in different FIGO stages.

Group			CCL22^low^ FOXP3^low^		CCL22^high^ FOXP3^low^		CCL22^low^ FOXP3^high^		CCL22^high^ FOXP3^high^	
			X^2^	*p*-Value	X^2^	*p*-Value	X^2^	*p*-Value	X^2^	*p*-Value
I–IIa	Log Rank (Mantel–Cox)	CCL22^low^ FOXP3^low^	−	−	10.547	0.001 *	0.263	0.608	6.107	0.013 *
		CCL22^high^ FOXP3^low^	10.547	0.001 *	−	−	3.122	0.077	0.010	0.920
		CCL22^low^ FOXP3^high^	0.263	0.608	3.122	0.077	−	−	3.000	0.083
		CCL22^high^ FOXP3^high^	6.107	0.013 *	0.010	0.920	3.000	0.083	−	−
IIb-IV	Log Rank (Mantel–Cox)	CCL22^low^ FOXP3^low^	−	−	2.575	0.109	0.013	0.911	4.761	0.029 *
		CCL22^high^ FOXP3^low^	2.575	0.109	−	−	0.896	0.344	0.119	0.730
		CCL22^low^ FOXP3^high^	0.013	0.911	0.896	0.344	−	−	1.891	0.169
		CCL22^high^ FOXP3^high^	4.761	0.029 *	0.119	0.730	1.891	0.169	−	−

* *p* < 0.05.

**Table 5 cancers-11-02004-t005:** Pairwise comparison of progression-free survival rate in different FIGO stages.

Group			CCL22^low^ FOXP3^low^		CCL22^high^ FOXP3^low^		CCL22^low^ FOXP3^high^		CCL22^high^ FOXP3^high^	
			X^2^	*p*-Value	X^2^	*p*-Value	X^2^	*p*-Value	X^2^	*p*-Value
I–IIa	Log Rank (Mantel–Cox)	CCL22^low^ FOXP3^low^	−	−	1.967	0.161	0.301	0.583	1.413	0.235
		CCL22^high^ FOXP3^low^	1.967	0.161	−	−	2.888	0.089	0.029	0.864
		CCL22^low^ FOXP3^high^	0.301	0.583	2.888	0.089	−	−	3.000	0.083
		CCL22^high^ FOXP3^high^	1.413	0.235	0.029	0.864	3.000	0.083	−	−
IIb–IV	Log Rank (Mantel–Cox)	CCL22^low^ FOXP3^low^	−	−	0.603	0.437	0.004	0.950	2.515	0.113
		CCL22^high^ FOXP3^low^	0.603	0.437	−	−	0.367	0.544	0.119	0.730
		CCL22^low^ FOXP3^high^	0.004	0.950	0.367	0.544	−	−	1.160	0.282
		CCL22^high^ FOXP3^high^	2.515	0.113	0.119	0.730	1.160	0.282	−	−

**Table 6 cancers-11-02004-t006:** Pairwise comparison of overall survival rate in different disease subtypes.

Group			CCL22^low^ FOXP3^low^		CCL22^high^ FOXP3^low^		CCL22^low^ FOXP3^high^		CCL22^high^ FOXP3^high^	
			X^2^	*p*-Value	X^2^	*p*-Value	X^2^	*p*-Value	X^2^	*p*-Value
Squamous carcinoma	Log Rank (Mantel–Cox)	CCL22^low^ FOXP3^low^	−	−	2.133	0.144	0.011	0.917	5.399	0.020
		CCL22^high^ FOXP3^low^	2.133	0.144	−	−	1.387	0.239	0.208	0.648
		CCL22^low^ FOXP3^high^	0.011	0.917	1.387	0.239	−	−	5.139	0.023 *
		CCL22^high^ FOXP3^high^	5.399	0.020 *	0.208	0.648	5.139	0.023	−	−
Adenocarcinoma	Log Rank (Mantel–Cox)	CCL22^low^ FOXP3^low^	−	−	1.802	0.179	1.363	0.243	4.093	0.043 *
		CCL22^high^ FOXP3^low^	1.802	0.179	−	−	0.264	0.608	1.824	0.177
		CCL22^low^ FOXP3^high^	1.363	0.243	0.264	0.608	−	−	1.000	0.317
		CCL22^high^ FOXP3^high^	4.093	0.043 *	1.824	0.177	1.000	0.317	−	−

* *p* < 0.05.

**Table 7 cancers-11-02004-t007:** Pairwise comparison of progression-free survival in different disease subtypes.

Group			CCL22^low^ FOXP3^low^		CCL22^high^ FOXP3^low^		CCL22^low^ FOXP3^high^		CCL22^high^ FOXP3^high^	
			X^2^	*p*-Value	X^2^	*p*-Value	X^2^	*p*-Value	X^2^	*p*-Value
Squamous carcinoma	Log Rank (Mantel–Cox)	CCL22^low^ FOXP3^low^	−	−	0.011	0.918	0.739	0.390	1.327	0.249
		CCL22^high^ FOXP3^low^	0.011	0.918	−	−	0.687	0.407	0.246	0.620
		CCL22^low^ FOXP3^high^	0.739	0.390	0.687	0.407	−	−	3.438	0.064
		CCL22^high^ FOXP3^high^	1.327	0.249	0.246	0.620	3.438	0.064	−	−
Adenocarcinoma	Log Rank (Mantel–Cox)	CCL22^low^ FOXP3^low^	−	−	0.770	0.380	0.796	0.372	4.175	0.041 *
		CCL22^high^ FOXP3^low^	0.770	0.380	−	−	0.264	0.608	1.824	0.177
		CCL22^low^ FOXP3^high^	0.796	0.372	0.264	0.608	−	−	1.000	0.317
		CCL22^high^ FOXP3^high^	4.175	0.041 *	1.824	0.177	1.000	0.317	−	−

* *p* < 0.05.

**Table 8 cancers-11-02004-t008:** Univariate and multivariate Cox regression analysis.

Clinicopathological Variables	Univariate Analysis		Multivariate Analysis		
	PFS	OS	PFS	OS (model 1)	OS (model 2)
**Age (≤50 years vs. >50 years)**					
HR	1.384	1.804	-	-	-
95%CI	0.804–2.385	0.897–3.629	-	-	-
*p*	0.241	0.098	-	-	-
**Tumor Size (≤4 cm vs. >4 cm)**					
HR	3.063	4.188	2.652	5.564	5.487
95%CI	1.710–5.486	1.924–9.112	1.063–6.617	1.538–20.130	1.504–20.011
*p*	0.001 *	0.001 *	0.037 *	0.009 *	0.010 *
**PN (Without Lymph Node Metastasis vs. Lymph Lode Metastasis)**					
HR	1.851	2.355	1.555	2.609	2.547
95%CI	1.065–3.218	1.162–4.774	0.756–3.200	1.032–6.596	0.988–6.562
*p*	0.029 *	0.017 *	0.231	0.043 *	0.053
**PM (Without Metastasis vs. Metastasis)**					
HR	1.710	3.303	-	-	-
95%CI	0.531–5.508	0.999–10.924	-	-	-
*p*	0.369	0.050	-	-	-
**FIGO (I–IIa vs. IIb–IV)**					
HR	2.630	2.913	1.088	0.584	0.599
95%CI	1.439–4.808	1.340–6.332	0.360–3.288	0.134–2.536	0.136–2.633
*p*	0.002 *	0.007 *	0.881	0.473	0.498
**Grade (I vs. II–III)**					
HR	1.672	1.054	-	-	-
95%CI	0.520–5.375	0.320–3.472	-	-	-
*p*	0.388	0.931	-	-	-
**Disease Subtype (Squamous Carcinoma vs. Adenocarcinoma)**					
HR	1.901	2.882	1.918	2.824	2.830
95%CI	1.043–3.466	1.408–5.899	1.038–3.545	1.359–5.869	1.361–5.883
*p*	0.036 *	0.004 *	0.038 *	0.005 *	0.005 *
**Number of CCL22+ cells (CCL22low vs. CCL22high)**					
HR	-	3.41	-	4.985	-
95%CI	-	1.567–7.419	-	2.206–11.266	-
*p*	-	0.002 *	-	0.0001 *	-
**Number of CCL22+FOXP3+ cells (CCL22lowFOXP3low vs. CCL22highFOXP3high)**					
HR	2.806	5.355	3.018	-	5.284
95%CI	0.864–9.115	1.580–18.154	0.923–9.869	-	1.513–18.456
*p*	0.086	0.007 *	0.068	-	0.009 *

* *p* < 0.05

**Table 9 cancers-11-02004-t009:** Antibodies used for immunohistochemical characterization and double immunofluorescence of cervical cancer samples.

Antibody	Isotype	Clone	Dilution	Source
CCL22	rabbit IgG	polyclonal	1:400 in PBS ^a^ 1:400 in Dako ^b^	Perprotech; DAKO(S322); Carpentera, CA, USA
FoxP3	mouse IgG	monoklonal	1:300 in PBS ^a^	Abcam
CD68	mouse IgG	polyclonal	1:1000 in PBS ^a^ 1:1000 in Dako ^b^	Sigma; DAKO(S322); Carpentera, CA, USA
CD163	mouse IgG	monoklonal	1:800 in PBS ^a^ 1:800 in Dako ^b^	Abcam; DAKO(S322); Carpentera, CA, USA
Cy-3 ^b^	goat IgG anti-rabbit	polyclonal	1:500 ^b^ in Dako ^b^	Dianova, Hamburg, Germany
Cy-2 ^b^	goat IgG anti-mouse	polyclonal	1:100 ^b^ in Dako ^b^	Dianova, Hamburg, Germany

^a^ antibodies used for immunohistochemistry, ^b^ antibodies used for immunofluorescence.

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
