# Peer review of "Higher CCL22+ Cell Infiltration is Associated with Poor Prognosis in Cervical Cancer Patients"

_cancers, 2019, doi:10.3390/cancers11122004_

Round 1
Reviewer 1 Report
The authors have very clearly improved the clarity and data of their article.
The analysis of public data confirms the results obtained in IHC. I just think it would be nice to keep the anlayses presented in the first version of the paper on public data between healthy and cancerous tissues indicating an increase in the expression of CCL22.
In my opinion, analysis on isolated monocytes with Caski/Hela cells is not the best way to explore relationships between tumor cells and macrophages it's nevertheless a first approach to confirm mechanism of CCL22 induction.
For the rest I have no comment to add.
Article can be accepted for publication after addition of data on healthy/cancerous tissue in the first figure.
Author Response
Reviewer 1:
Comments and suggestions for authors: The authors have very clearly improved the clarity and data of their article.
The analysis of public data confirms the results obtained in IHC. I just think it would be nice to keep the analyses presented in the first version of the paper on public data between healthy and cancerous tissues indicating an increase in the expression of CCL22.
In my opinion, analysis on isolated monocytes with Caski/Hela cells is not the best way to explore relationships between tumor cells and macrophages it's nevertheless a first approach to confirm mechanism of CCL22 induction.
For the rest I have no comment to add.
Article can be accepted for publication after addition of data on healthy/cancerous tissue in the first figure.
Reply to minor point 1: Thank you for your suggestion. We have replaced the figure 1 with the picture in the first version. In addition, to enhance the richness of the paper, we also keep the picture in the second version as the supplementary materials in PDF.
The picture is as followed:
Fig. 1: Transcripts expression level of CCL22 in CESC tissue explored by GEPIA database. Red and grey colors denote the expression level in tumor tissue and normal tissue, respectively. CESC, cervical squamous cell carcinoma and endocervical adenocarcinoma
Special thanks to you for your comments!

Reviewer 2 Report
The authors responded adequately to all our suggestions.
Author Response
Reviewer 2: The authors responded adequately to all our suggestions.
Reply to reviewer 2: We appreciate your recognition very much!
Once again, thank you very much for your comments and suggestions!

This manuscript is a resubmission of an earlier submission. The following is a list of the peer review reports and author responses from that submission.
Round 1
Reviewer 1 Report
Qun Wang et al's article aims to define the impact of the CCL22 chemokine on the survival of patients with cervical cancer. It shows that CCL22 is associated with lower survival and appears to be associated with the presence of regulatory T cells. This could be explained by the expression of CCR4 on the surface of Treg cells. The authors wish to show that CCL22 is produced by intratumoral macrophages (TAM) and that they are therefore responsible for the recruitment of Treg.
The theme of this article is relevant and could in the future offer new therapeutic approaches combining antibodies targeting immune checkpoints and inhibitors of the CCR4 / CCL22 axis or TAM2 in cervical cancer. This is an interesting study since it presents results on a large cohort of cervical cancer by IHC, while trying to address mechanistic questions about the origin and impact of CCL22 in tumor tissue immunology. Nevertheless, the article cannot be accepted as it stands and some points need to be improved to make it more complete.
Minor points:
The introduction needs to be improved to bring more binding between the context of cervical cancer, CCL22, Treg and macrophages. Introduction is too much in the form of a listing.Quality of figure 3 is not sufficient. The magnification box does not indicate where the zoomed area is. Foxp3 staining is not clear and picture are blurry.
The table 1 is not a correlation analysis. Title of the table have to be change.
In table 4: change “monoklonal” by “monoclonal”.
Major points:
In figure 1, CCL22 gene expression is analyzed in cervical cancer and normal tissue using GEPIA database. It’s a good point to use this type of data. The authors should compare the expression of metagen associated with Treg, TAM2 and CTL in cervical cancer versus normal tissue. Are they different? Are they positively or negatively correlated with CCL22? Does the prognostic role of CTL is impacted by absence or presence of CCL22? Authors can use Treg, CTL or TAM2 signatures described in this article Bindea G et al, 2013, PMID 24138885). Showing by this type of approach that CCL22 is well associated with an immunosuppressive profile (Treg / TAM2) and a lower CTL response in cervical cancer could greatly improve the article and strengthen the data obtained with IHC analysis.In table 1, a total of 250 patients is indicated. If we look at the number of patient in IRS low or high, we have respectively 144 and 103 patients, the total is not 250. Why? Does this correspond to missing data in the database? If yes, it’s important to indicate this information. It’s also the case for others parameters analyzed. This point complicates the reading of the table.
In the table 1, HPV status is not presented. This point is essential in a context of cervical cancer. Moreover, HPV status can have an impact on parameters studied, particularly CCL22 and Foxp3 (Zhao M et al, Virol J, 2017, PMID: 28086903). With this information, authors have to analyzed CCL22 and Foxp3 expression in HPV positive versus negative tumors. If the HPV status modulates the expression of CCL22, Foxp3, this could bring new elements on the immunotherapies strategies to adopt in cervical cancer.
In the table 1 authors have to add therapeutic strategies used for CC patients.
In figure 2, patients overall survival is analyzed in three groups. Why this choice? It’s not more correct to have :
- CCL22low/Foxp3low
- CCL22high/Foxp3low
- CCL22low/Foxp3high
- CCL22high/Foxp3high
Please add under survival curve the numbers at risk. PFS have to be analyzed.
It could be interesting to analyze the impact of CCL22/Fopx3 on PFS and OS in low and in high FIGO stages or grade. Indeed, therapeutic strategies, CCL22 and Foxp3 expression are not the same in I-Ib2 and in II-IV stages or grade.
Figure 6: The link between TAM2 and CCL22 is explore with CCL22, CD68 and CD163 co-staining by IF-IHC on TMA. This strategy is interesting because we can have directly the source of CCL22 in tumor microenvironment.
Authors have to present an isotype control primary antibody for CCL22, CD68 and CD163 to check antibody specificity. Scale is absent in picture. Please present also an HES view of semi-serial section to localize if CCL22+/CD163+ cells are present in tumor core, in invasive margin, ect. How many tumors were analyzed? Please, perform a statistical analysis of the number of CD68+/CCL22-, CD68+/CCL22+ and CD163+/CCL22-, CD163+/CCL22+ cells in a representative number of tumor samples. This analysis is essential to validate clearly the link between TAM2 and CCL22.
Reviewer 2 Report
This is an interesting retrospective study but it is only an observational analysis.
The authors highlight a correlation between CCL22 and FoxP3 expression in cervical cancer and a strong relation between CCL22 and prognosis.
Some points seem to be lacking in the study:
First of all the study is carried out on tissue samples included in a Tissue Micro Array. The main limitation of this method is linked to its poor ability to represent the tumor microenvironment in its entirety. Since the analysis is oriented to the characterization of immune cells, validation on a whole section for selected cases would be necessary.
The most critical point is however the lack of functional studies aimed to understand the mechanisms leading to CCL22 expression by tumor-infiltrating immune cells. For example, to test whether CCL22 secretion may be induced by the tumor cells could be realized co-cultures of human peripheral blood mononuclear cells co-incubated with different human cervical cancer cell lines etc. Co-culture with tumor cells strongly increased CCL22 levels of PBMC in different human cancer (Wiedemann GM et al Cancer cell-derived IL-1α induces CCL22 and the recruitment of regulatory T cells. 2016 Apr 25;5(9):e1175794.)
The study does not appear to be adequate for the journal's targets.
Reviewer 3 Report
In this study, authors investigated the role of CCL22 in clinical cases of cervical cancer. Tissue microarray technique has been applied to identify a correlation between the expression of CCL22 and FoxP3 at the protein level, supporting that CCL22 expressing cells, both tumor-derived and infiltrating, act in recruiting regulatory T-cells. Immunostainings against CD68 and CD163 demonstrated that CCL22 is produced by M2-like macrophages infiltrating the tumor. Authors discuss the prognostic value of CCL22 as a biomarker. The manuscript presents valuable findings, nonetheless, some issues need to be addressed:
1- The abstract is missing a rationale/background/hypothesis/objective of the study.
2- In table 3, what is it meant by “PT” and “PN”? Is this related to the pTNM staging?
3- On Figure 3, can authors provide images of improved quality at least for FoxP3 (more) panel?
4- Can authors elaborate on the impact of this finding on clinical management/treatment of cervical cancer?